# Water-Vapour Monitoring from Ground-Based GNSS Observations in Northwestern Argentina

**Nikolaos Antonoglou** [1,2,*] , **Kyriakos Balidakis** [2] , **Jens Wickert** [2,3] , **Galina Dick** [2] , **Alejandro de la Torre** [4] **and Bodo Bookhagen** [1]

1   Institute of Geosciences, University of Potsdam, Karl-Liebknecht-Str. 24-25, 14476 Potsdam, Germany
2   German Research Centre for Geosciences GFZ, Telegrafenberg, 14473 Potsdam, Germany
3   Institute of Geodesy and Geoinformation Science, Technische Universität Berlin, Straße des 17. Juni 135, 10623 Berlin, Germany
4   Facultad de Ingeniería, Universidad Austral, Mariano Acosta 1611, Pilar B1630, Argentina
*   Correspondence: nikolaos.antonoglou@gfz-potsdam.de

**Abstract:** The Central Andes in northwestern Argentina are characterized by steep topographic and climatic gradients. The humid foreland areas at 1 km asl elevation rapidly rise to over 5 km in the eastern Cordillera, and they form an orographic rainfall barrier on the eastern windward side. This topographic setting combined with seasonal moisture transport through the South American monsoon system leads to intense rainstorms with cascading effects such as landsliding and flooding. In order to better quantify the dynamics of water vapour transport, we use high-temporal-resolution global navigation satellite system (GNSS) remote sensing techniques. We are particularly interested in better understanding the dynamics of high-magnitude storms with high water vapour amounts that have destructive effects on human infrastructure. We used an existing GNSS station network with 12 years of time series data, and we installed two new ground stations along the climatic gradient and collected GNSS time series data for three years. For several stations we calculated the GNSS signal delay gradient to determine water vapour transport direction. Our statistical analysis combines in situ rainfall measurements and ERA5 reanalysis data to reveal the water vapour transport mechanism for the study area. The results show a strong relationship between altitude and the water vapour content, as well as between the transportation pathways and the topography.

**Keywords:** GNSS meteorology; GNSS remote sensing; intense rain events; water vapour; Central Andes; orographic barrier; South American monsoon system

## 1. Introduction

Strong rainfall events repeatedly lead to natural disasters in steep mountain regions, e.g., [1–5]. Especially along the eastern Andes, intense hydro-meteorological events cause landsliding and flooding that impact population and infrastructures [6–8]. Previous studies indicate that hydro-meteorological extreme events are often a consequence of several additive climatic and topographic factors [9]. For example, the availability of high water vapour transported through the South American low-level jet (SALLJ) from the north along the eastern Andes, the advancement of cold fronts from the south, and the steep topography lead to unstable atmospheric conditions and heavy cloudbursts in the eastern Central Andes [4,6].

A core component for understanding these hydrometeorologic processes are highly dynamic observations. While satellite-based observations have advanced our understanding of large-scale orographic effects at windward sides of mountain ranges, e.g., [10,11], these data often do not provide the temporal resolution to decipher dynamic processes, including the vertical and horizontal components of water vapour transport within storms. Similarly, reanalysis and numerical weather prediction data are very useful for understanding the dynamics of large-scale processes, but they often do not allow for the reliable measurement

of spatially small (<2 km) and temporally short (<1 h) processes, e.g., [9,12]. With the recent advancement of global navigation satellite system (GNSS)-based observation and the increase in availability of these data, an alternative meteorological observation method is readily available, e.g., [13–15].

The study area in northwestern Argentina is on the eastern side of the second largest orogenic plateau—the Altiplano–Puna plateau. The plateau is rich in mineral and georesources, especially lithium. Maintaining infrastructural networks in this area is important, but the seasonal South American monsoon system (SAMS) heavily impacts the road network connecting the high-elevation and resource-rich areas with the low-elevation foreland areas. In addition, global warming leads to changing rainfall and discharge patterns, shifting the distributions of low frequency and high magnitude hydrometeorological events [4,9,16,17].

In this study, we explore the potential of GNSS-based observations to track water vapour transport along and across the south-central Andes. For this purpose, we used available station data since 2010, but also installed additional monitoring sites in medium and high elevation areas for the duration of three years (cf. Figure 1). We use GNSS time series data to group stations into clusters and use their high temporal resolution to analyse water vapour distributions. Furthermore, we implement an analysis of water vapour gradients using GNSS delays to determine direction of water vapour transport. Our investigation is complemented by in situ weather stations and ERA5 reanalysis data to validate results and provide a hydrometeorological context.

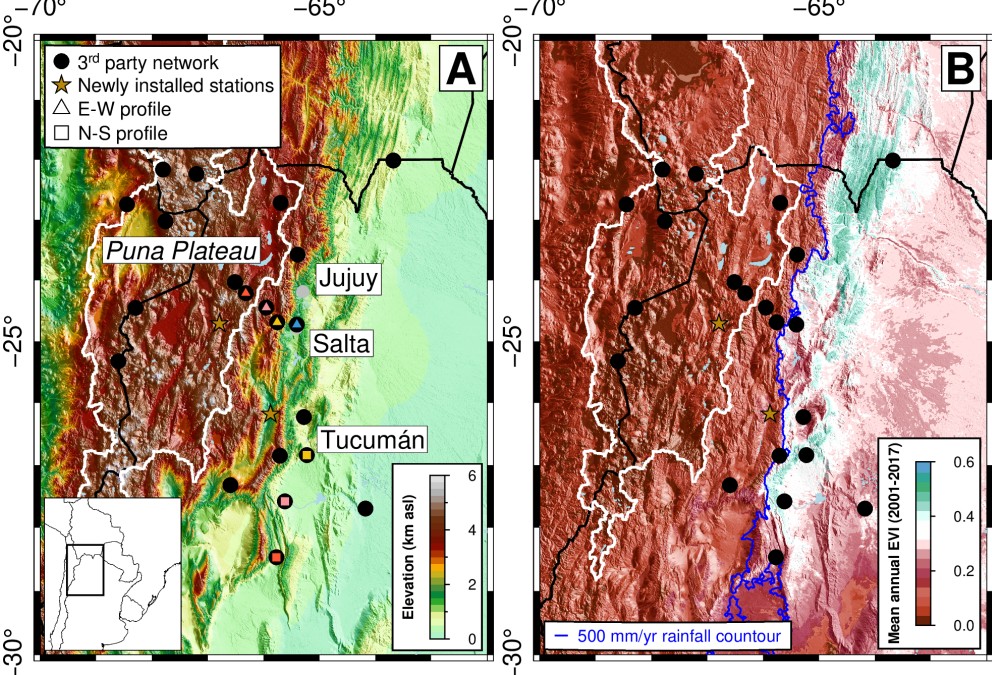

**Figure 1.** Topographic setting of the south-central Andes with GNSS station network location. We have selected a subset of these stations for an analysis of water vapour transport: across the Andes in east–west (E-W) direction (UNSA, GOLG, SRSA, and SALC) and along the Andes in north–south (N-S) direction (UNSA, TUCU, JBAL, and CATA). UNSA, GOLG/TUCU, SRSA/JBAL, and SALC/CATA are represented with blue, yellow, pink, and orange colours, respectively. The white line outlines the internally drained Altiplano–Puna plateau, also called the Central Andean plateau. Black lines are international borders (**A**). (**B**) shows enhanced vegetation index (EVI) information for the area of interest between 2001 and 2017. The blue line points out 500 mm annual rainfall (Topographic data obtained from ETOPO1 [18], EVI data obtained from MODIS/Terra [19], rainfall information retrieved from the Tropical Rainfall Measuring Mission (TRMM) [20], as cited in Bookhagen and Strecker [10]).

## 2. Climatic Setting of the Study Region in Northwestern Argentina

The climatic setting of the study area in the Central Andes in northwestern Argentina is controlled by several interfering factors. In short, these are: (1) the moisture transport from tropical regions; (2) unstable atmospheric conditions at the boundary between tropical and subtropical air masses; (3) the steep topographic gradient leading to orographic processes; and (4) cold temperature excursion from the southern polar region. This leads to strong climatic gradients in the east–west (E-W) and north–south (N-S) directions.

The conveyor belt transporting water vapour from north to south east of the Andes is the South American low-level jet (SALLJ). The authors in Montini et al. [21] showed that moisture transport associated with the jet is greater during summer, when the SALLJ is influenced by warm, moisture-rich air masses from tropical South America. During the austral summer (DJF), it mainly transports moist air from the latitudes of the Amazon southward [22,23]. Reanalysis data show that the inter-annual variability of the jet's strength and frequency is significantly modulated by the El Niño southern oscillation, especially during spring [24].

Moreover, the SALLJ is controlled by the dynamics of the SAMS. From a synoptic and seasonal perspective, the SAMS is highly dependent on the surface temperature difference between the ocean and surrounding land masses [23,25,26]. During the SAMS activity in the DJF, more than 50% of annual rainfall occurs along the Andes. The same region receives less than 10% of the annual rainfall in JJA season, which leads to distinct warm-wet and cold-dry seasons, e.g., [17,23].

The Central Andes in northwestern Argentina are located near the end of the SALLJ and also receive moisture from mesoscale convective systems [27–29]. To the east of the study area exists one of the global hotspots for mesoscale convective systems. The complex topography and interaction of unstable air masses leads to some of the heaviest rainfall and largest cloudbursts on the South American continent, e.g., [4,6,16,30]. An additional important component influencing heavy cloudbursts in this area is the interplay of high moisture availability through the SALLJ and cold surges and frontal systems propagating from the south [4,29]. This is an important process, especially in the generation of areal-extensive heavy rainfall leading to major flooding in the eastern Central Andes and downstream areas. Previous studies have shown that 80% of the 40 largest discharge events of the past 40 years are associated with propagating cold fronts [4]. The interplay of these factors lead to complex rainfall patterns on the South American continent and in the study area [31].

In addition to the atmospheric conditions, the topographic setting of the study area with deeply incised valleys funneling moisture to the higher-elevation area lead to pronounced orographic barriers on the eastern, windward sides of the mountain ranges [10,15,32]. The authors of Castino et al. [17] suggest three climate zones that follow the topographic and rainfall gradient from east to west: a low-elevation and low-slope sector, a medium-elevation sector dissected by rivers with steep hill slopes, and a high-elevation sector with moderate to steep slopes. The climatic contrast between these areas is large: the foreland zone receives more than 1.5 m/yr rainfall, whereas the high-elevation areas receive less than 0.2 m/yr [10]. The intermediate elevations show high topographic relief and force orographic lifting and convection resulting in pronounced rainfall on the windward slopes [6,9,33]. The convective cells have lifetimes of several hours and observations with high temporal resolution are required [23,34,35]. In addition to the spatial and inter-annual rainfall distribution patterns, there is also an inter-diurnal variation. According to meteorological data from Salta [36], we observe a nocturnal rainfall peak during DJF between 21:00 and 4:00 local time [37].

## 3. Data

We processed GNSS and reanalysis data from ERA5 [38], and our analysis focuses on the years between 2010 and 2021 [39]. This time frame is mainly constrained by the availability of the GNSS data.

### 3.1. GNSS Network Description

We analysed data from 23 GNSS stations listed in Table 1. A total of 16 stations are located in northwestern Argentina, four stations in Chile, and three stations in Bolivia. The elevation range of the station spans more than 5 km, and the network extends 450 km in the E-W and 700 km in the N-S direction. This area is impacted by the moisture transport associated with the SALLJ from north to south, but also the westward moisture transport across the orographic barrier.

**Table 1.** Geographic coordinates of the stations that were used for water vapour analysis. The institutions that were responsible for the installation of the facilities are listed in the column **Source** and are the National Geographic Institute of Argentina (Instituto Geográfico Nacional—IGN), UNAVCO, the University of Potsdam (UP), and the German Research Centre for Geosciences (Deutsches GeoForschungsZentrum—GFZ). The column **Analysis Centre** shows where the data processing was carried out: either at the GFZ or the Nevada Geodetic Laboratory (NGL).

| Station Name | Latitude | Longitude | Height (m) | Source | Analysis Centre |
|---|---|---|---|---|---|
| ABRA | 22°43′19.32″S | 65°41′50.31″W | 3530.10 | IGN | NGL |
| ALUM | 27°19′24.33″S | 66°35′47.86″W | 2736.94 | IGN | NGL |
| CAFJ | 26°10′51.22″S | 65°52′49.17″W | 1702.36 | UP/GFZ | GFZ |
| CATA | 28°28′15.54″S | 65°46′26.83″W | 547.15 | IGN | NGL |
| CBAA | 22°44′46.92″S | 68°26′53.33″W | 3514.84 | UNAVCO | NGL |
| CJNT | 23°01′38.96″S | 67°45′38.06″W | 5074.05 | UNAVCO | NGL |
| COLO | 22°10′02.57″S | 67°48′14.32″W | 4376.93 | UNAVCO | NGL |
| GOLG | 24°41′26.11″S | 65°45′38.80″W | 2381.15 | UNAVCO | NGL |
| JBAL | 27°35′03.86″S | 65°37′21.89″W | 409.16 | IGN | NGL |
| LCEN | 25°19′33.81″S | 68°36′09.36″W | 4270.94 | UNAVCO | NGL |
| PUNJ | 24°42′46.96″S | 66°47′37.27″W | 3802.58 | UP/GFZ | GFZ |
| SALC | 24°12′47.11″S | 66°19′20.83″W | 3841.62 | UNAVCO | NGL |
| SOCM | 24°27′16.60″S | 68°17′42.59″W | 3969.45 | UNAVCO | NGL |
| SRSA | 24°26′59.24″S | 65°57′11.85″W | 3153.80 | UNAVCO | NGL |
| TAVA | 26°51′10.72″S | 65°42′36.02″W | 2036.74 | IGN | NGL |
| TERO | 27°41′57.30″S | 64°10′42.17″W | 222.63 | IGN | NGL |
| TIL2 | 23°34′37.70″S | 65°23′42.26″W | 2517.78 | IGN | NGL |
| TRNC | 26°13′48.77″S | 65°16′55.82″W | 816.08 | IGN | NGL |
| TUCU | 26°50′35.71″S | 65°13′49.26″W | 485.02 | IGN | NGL |
| TUZG | 24°01′53.82″S | 66°30′59.56″W | 4338.67 | UNAVCO | NGL |
| UNSA | 24°43′38.84″S | 65°24′27.51″W | 1257.79 | IGN | NGL |
| UTUR | 22°14′31.21″S | 67°12′19.94″W | 5184.09 | UNAVCO | NGL |
| YCBA | 22°01′01.56″S | 63°40′47.94″W | 659.66 | IGN | NGL |

The station data are maintained by different data providers, and they have been compiled and prepared for this study. The stations CAFJ and PUNJ were installed by the University of Potsdam (UP) and the German Research Centre for Geosciences (Deutsches GeoForschungsZentrum—GFZ), while their observations were processed by the GFZ. All other stations were installed by UNAVCO [40–52] and by the National Geographic Institute of Argentina (Instituto Geográfico Nacional—IGN) [53]. The Nevada Geodetic Laboratory (NGL) provides access to these data [54].

Due to the heterogeneous development of the GNSS network, observations do not always overlap. Generally, the network spans the years from 2010 to 2021, but only six time series extend through the entire range. Furthermore, all stations have undergone data loss for limited periods of time due to technical reasons.

For the analysis of the topographic impact, we selected four stations across the central Andes in an E-W direction (UNSA, GOLG, SRSA, and SALC, cf. Figure 1). Moreover, four stations were chosen (UNSA, TUCU, JBAL, and CATA) perpendicular to this and along a N-S direction. Those two subsets have one station in common, they cover multiple years with simultaneous observations, and they reflect the diverse climate conditions of the region. The largest elevation difference exists between the stations UNSA at 1224 m and SALC at

3799 m asl. In this research, we use these two stations to assess data measurements from different climatic conditions. In addition, we rely on the stations UNSA, CAFJ, and PUNJ for in situ comparisons because these are accompanied by rain gauge sensors in short distance. The water vapour readings in those locations is directly related to high-precision rainfall information.

### 3.2. ERA5 Data

We rely on ERA5 hourly data on pressure levels from 2010 to 2021 [38], and we use the native temporal and spatial resolution (0.25°). ERA5 data span several decades, and they are continually updated with a minimal latency of a few days. Furthermore, ERA5 analyses 37 pressure levels, reaching a height of 80 km. This way we can calculate refraction and wind speed information, which is vital for the methodology that we follow. Finally, we linearly resample ERA5 data (with a temporal resolution of one hour) to the 5 min temporal resolution of the GNSS data.

## 4. GNSS Meteorology

GNSS meteorology is a methodology for acquiring neutral atmosphere information by employing GNSS measurements. The primary output of this technique is the slant total delay between the receiver and the satellite. In this section, we discuss the translation of this product into the zenith total delay and its gradients along the E-W and N-S directions, and subsequently, the calculation of the zenith hydrostatic delay, the zenith wet delay, and the water vapour amount. We also describe the approach to derive the gradients of the zenith hydrostatic and wet delay.

### 4.1. Slant Delay Decomposition

The atmospheric delays of the GNSS signals, both in the slant and the zenith directions, are composed of the hydrostatic (or dry) and the wet counterparts [55]. In order to decompose the slant total delay, we project these to the vertical using mapping functions. The simplest versions of the mapping functions assumes a uniform atmosphere and mainly depends on the elevation ($\varepsilon$), whereas the more advanced versions take into account azimuthal asymmetry [56,57]. The latter approximations yield significantly better results because they better reflect the reality by introducing gradients along the E-W and N-S directions. According to Kačmařík et al. [58], a complete expression of the observation equation of the slant total delay can be written as follows:

$$S_{total} = m_{dry}Z_{dry} + m_{wet}Z_{wet} + m_{grad}(G_{NS}\cos(a) + G_{EW}\sin(a)) \tag{1}$$

where:

| | |
|---|---|
| $a$ | azimuth |
| $Z_{dry}, Z_{wet}$ | zenith hydrostatic and wet delay |
| $m_{dry}, m_{wet}$ | mapping functions for the dry and wet component |
| $m_{grad}$ | mapping function of the gradient parts |
| $G_{NS}, G_{EW}$ | gradients in the N-S and E-W directions |

In a later step, the water vapour is directly calculated by the slant total delay. The formula for this conversion is implemented by Bevis et al. [59], and it can be written as follows:

$$WV = Z_{wet}\Pi \tag{2}$$

$$\Pi = \frac{10^6}{\varrho R_u[(k_3/T_m) + k_2']} \tag{3}$$

$$k_2' = k_2 - mK_1 \tag{4}$$

where:
$\varrho$          density of liquid water
$R_u$          specific gas constant of water
$m$          ratio of molar masses of water vapour and air
$k_1, k_2, k_3$    physical constants

### 4.2. Data Processing

Because the GNSS data were not obtained from a homogenized network (cf. Section 3.1), various software packages are used for the calculation of the zenith total delay and its gradients. More specifically, Receiver Independent Exchange Format files are processed with Earth Parameter and Orbit System (EPOS) [60] and GipsyX [61] software packages. Although both programs are very robust and they rely on the same fundamentals, each application follows its own strategy, and it is important to know the detailed differences for a proper assessment of the analysis results.

#### 4.2.1. EPOS

EPOS is a GNSS analysis software that was developed in the 1990s by the scientific team of the GFZ [60]. Even though it was initially designed for space applications (e.g., precise orbit determination), it can be also used for terrestrial applications. More specifically, EPOS estimates the slant total delay in near real-time mode using precise point positioning algorithms [62,63]. For each set of epochs, the zenith hydrostatic and wet delay, and the gradients of the zenith total delay are estimated in a least squares adjustment where the functional model is Equation (1). Additionally, the utilized mapping function for the zenith components is the global mapping function [64], while for the azimuthal component it is a mapping function described by Bar-Sever et al. [57].

#### 4.2.2. GipsyX

Gipsy is a multi-purpose navigation software that was developed in the 1980s by the Jet Propulsion Laboratory [61]. Its last version is GipsyX and it does not only allow for GNSS data processing, but for other space geodetic techniques, such as satellite laser ranging and Doppler orbitography and radiopositioning integrated by satellite. The processing strategy of the GNSS data is similar to EPOS. The only difference is that GipsyX employs the Vienna Mapping Function 1 [65,66] instead of a global mapping function [54,61].

### 4.3. Ray Tracing

To obtain a signal delay while propagating through the atmosphere, we adopted the geometrical optics approximation. First, we calculated the index of refraction based on hourly ERA5 data on equiangular 0.25° grids up to the maximum height of 80 km; we rely on pressure, geopotential height, temperature, and specific humidity of all ERA5 levels.

To perform ray tracing, we adopt a variational approach (Euler–Lagrange equations) employing an implicit finite difference scheme [67–69]. The atmospheric delays are calculated by integrating refractivity along the ray path. The ray-traced delays describe how atmospheric delay varies with elevation and azimuth. Because it is not practical to employ geodetic observations, delay is often described as a continuous function of elevation and azimuth as shown in Equation (1). The weather model-derived zenith delays and gradients are comparable to those from space geodetic data analysis.

While zenith delays are not affected by the choice of the parametric model describing directional delay variations, the gradient components that describe azimuthal delay variations do. In particular, the choice of the scheme describing the elevational decay of the asymmetric delay with increasing elevation is crucial. Although the latest International Earth Rotation Service [70] recommends a first-order continued fraction form for $\sin(\varepsilon)\cot(\varepsilon)\left((\sin(\varepsilon)\tan(\varepsilon) + 0.0032)^{-1}\right)$ [56], it is popular to use a gradient mapping function that involves the wet mapping function ($m_{wet}\cot(\varepsilon)$) [57]. As demonstrated in Balidakis [71] and Kačmařík et al. [58], these two sets of gradients are incompatible, and they

cannot be accurately transformed to be used with other gradient mapping functions. Because the latter gradient mapping function has been adopted during the estimation of first-degree gradient components employing GipsyX [54] and EPOS [60], we adopt consistent elevation-decay modelling to estimate gradients from our ray-traced delays. That is, in the station- and epoch-wise least squares adjustment of ray-traced delays, we constrained the estimates of the dry and wet symmetric mapping function to the values given by the gridded version of Vienna Mapping Function 1 so that we may directly compare ray-traced gradients with GNSS-derived gradients.

## 5. Analysis Methods

We performed spatio-temporal analyses by applying several methods: First, we identified temporal correlations between GNSS time series through k-means clustering. Second, we used spectral analysis to identify recurrence of specific event magnitudes. Third, we related GNSS-based water vapour observations with rainfall measurements for several in situ meteorologic stations, and we analysed the frequency–magnitude distributions of water vapour and rainfall. Furthermore, fourth, we determined latitude moisture transport through zenith delay gradient measurements.

### 5.1. k-Means Clustering of GNSS Time Series

We used k-Means clustering of monthly averaged water vapour measurements to group similar station data. More specifically, we employed the TimeSeriesKMeans algorithm within the Tslearn library [72], which is especially implemented for time series. Using this algorithm, we relied on Euclidean distances, because the seasonal meteorological of various locations data differ in amplitude and not in frequency, and they are not subject to time shifts. We set the number of clustering classes to three, because the topographic and climatic setting creates three zones in low-, medium-, and high-elevation areas [17] (cf. Figure 1 and Section 2).

Due to the lack of simultaneous and overlapping observations (cf. Figure 2), the clustering for all years cannot be realized. Instead, we selected only the measurements from 20 stations during 2014, when there were no major data interruptions. Even though this is a fraction of the available data, it is adequate for the classification because it covers an entire annual cycle, and the selected stations are distributed over various altitudes along the E-W and N-S cross-sections.

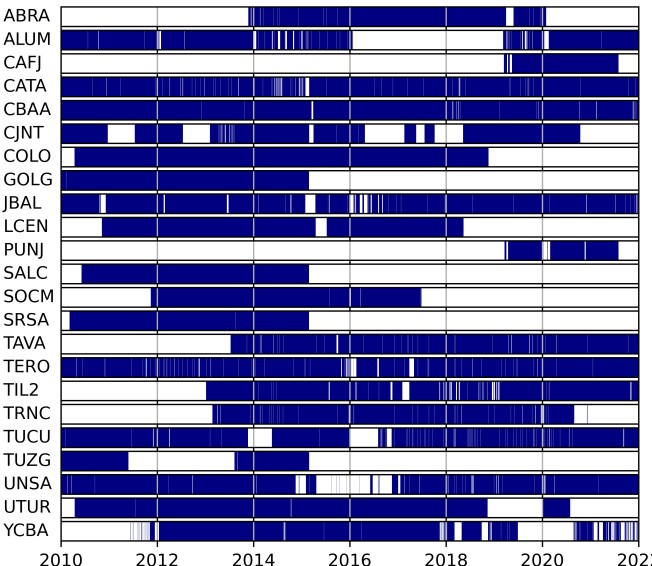

**Figure 2.** Temporal coverage of the stations used in this study. Blue areas indicate times with data availability.

### 5.2. Spectral Analysis

The GNSS water vapour signal was generated by overlapping periodic oscillations and we decomposed the signals by analysing their spectral behaviour. The annual time series depict the water vapour observations in a low- (UNSA) and a high-altitude (SALC) station during 2012 with a daily temporal resolution (Figure 3). In this characteristic overview, one can clearly identify the annual oscillation in both stations. Moreover, there are shorter oscillations with frequencies of about one week, which correspond to the synoptic-scale water vapour cycles in the atmosphere. We also show the 5 min water vapour time series illustrating the sub-daily cycles for two periods in summer and winter.

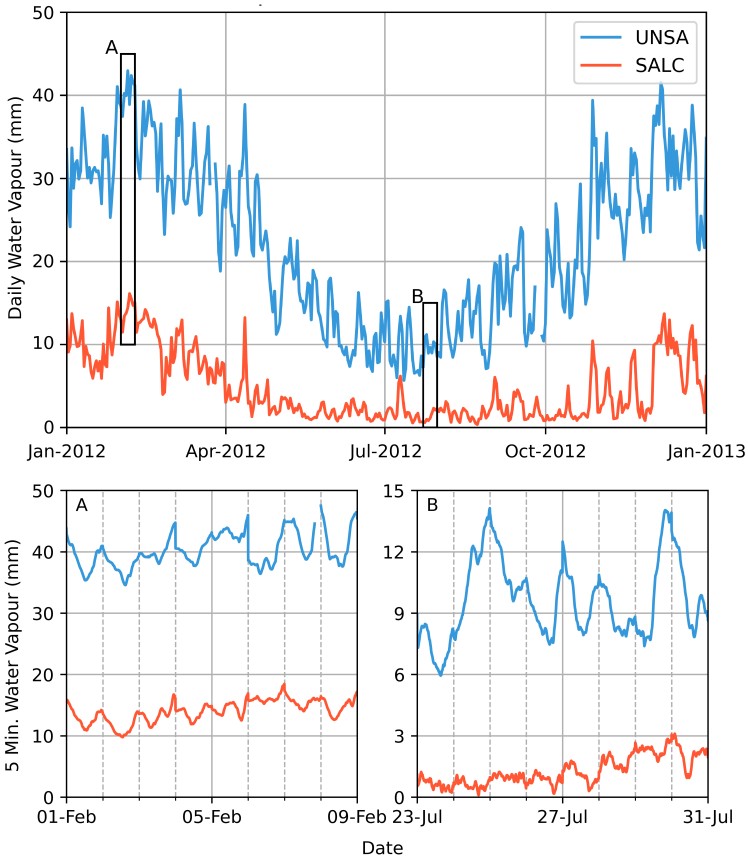

**Figure 3. Top:** Water vapour observations with 1 day sampling rate for the UNSA and SALC stations during 2012. **Bottom:** Detailed view with 1 h sampling rate during two incidents in February (**A**) and July (**B**) 2012. The panels have a different scales in their y-axes because of the contrasting water vapour levels during the wet and dry season and their different temporal resolutions.

The frequency domain analysis is accomplished by generating 3D graphs of the signal responses at various frequencies over time (or spectrograms) with a series of Fourier transforms [73,74]. Considering the non-simultaneous observations at all stations, we selected data between 2010 and 2014, when the stations along the E-W and N-S cross-sections were functional. We set the sampling window to seven days because it is aligned with the period of the synoptic events, and we forecast data discontinuities smaller than two weeks using the Prophet algorithm. This approximation employs an additive model that is sensitive to periodic fluctuations. It decomposes the signal into a trend, seasonality, and irregular occurrences, and it does not require regularlyspaced data as input [75].

For the next step, we quantified the spectral signals. Transient water vapour variations $y$ were approximated as the sum of four different signal groups:

$$y = y_p + y_h + y_a + y_n \tag{5}$$

where $y_p$, $y_h$, $y_a$, and $y_n$ indicate the polynomial term, the harmonic variations, the synoptic term, and the noise, respectively.

The polynomial counterpart ($y_p = \sum_i x_i t^i$) is time ($t$) dependent, and it yields largely non-significant estimates for coefficient terms $i > 1$. It will not be discussed herein given the relatively short data duration. The harmonic variations that are of main interest are described as follows:

$$y_h(t) = \sum_j A_j \cos(\chi_j(t) - \phi_j) \tag{6}$$

where $j$ is a certain wave with a particular frequency, $A_j$ denotes the amplitude, $\phi_j$ denotes the phase, and $\chi_j$ denotes the astronomical argument. To build the latter, we adopt Doodson multipliers from Hartmann and Wenzel [76]. We estimate in-phase and quadrature components for all waves whose speed spans from the Nyquist frequency to one cycle per half-time series length, making sure that the Rayleigh criterion is fulfilled for all possible pairs, while water vapour features marked modulation in certain spectral lines (e.g., $S_1$ and $S_2$) it is not explicitly considered, because the purpose of the this step is the estimation of the power spectral density (PSD) of the post-fit residual time series, and eventually to perform a scaling analysis using power-law fitting [77]. We refer the interested reader to Balidakis et al.[78] and the accompanying supplementary material for further details on the estimation of harmonic amplitudes from meteorological time series. We prefer this approach to adopting a bandstop filter (as in [79]) because the gap filling strategy may introduce artificial spectral signatures. A power-law distribution is suitable for describing natural phenomena because we assume that the probability of an event is inversely proportional to the power of its magnitude as it has been documented in several other natural science datasets, e.g., [80].

To estimate the PSD given the water vapour post-fit residuals, we utilize the multitaper method, e.g., [81]. Two sets of data are employed: raw post-tidal-fit residuals with their uncertainty estimates and a normalized version of the former by its standard deviation to facilitate the direct comparison of the PSD estimates from several data types stemming either from the weather model (ERA5) or the GNSS observations themselves. The motivation behind employing post-fit residuals instead of the raw water vapour time series is that the sharp spectral lines at frequencies associated with radiative forcing ($S_a$ and $S_1$, as well as overtones thereof) bias the estimation of the spectral indices in the power-law approximation of the PSD.

### 5.3. Water Vapour and Rainfall Relation

We analysed the relation between liquid precipitation and water vapour to better understand their relation. The magnitude of water vapour varies throughout the season and during rainstorm events, but water vapour is always present in the atmosphere. On the other hand, rainfall occurs during events or in the form of episodes, when water vapour reaches the peak relative humidity level (100%), and it forms water particles in various forms [82]. In an initial analysis, we selected all daily averaged water vapour values that exceed the 90th percentile and the corresponding daily summed rainfall, and we analysed their relationship in a power-law framework. This data subset also includes zero-rainfall days and we exclude those epochs from the power-law relation analysis. We repeated the analysis by selecting daily rainfall exceeding the 75th percentile and the associated daily averaged water vapour, and we examined the relationship of their linearly binned readings in a quantile–quantile (Q-Q) plot. For this comparison analysis, we selected the UNSA, CAFJ, and PUNJ stations for the period between August 2019 and July 2021. UNSA is a long-term station, and the other two sites were specifically installed for this study. All three stations are accompanied by in situ rain-gauges.

### 5.4. Latitudinal Moisture Gradient Transport

In order to better understand the dynamics of the moisture transported by the SALLJ and the impact of the strong orographic effect in our study area, we examined the zenith

wet delay gradients (or wet gradients). In order to detect correlation between wind vectors and wet gradients, we firstly plotted their azimuthal distributions. We removed potential wind shear effects with the surface by only considering pressure levels exceeding the altitude by 1 km above the surface. Subsequently, we focused on the variation of the wet gradients per epoch. We analysed the seasonal distributions and, apart from the number of events per azimuthal segment, we also showed the 90th to 50th percentile ratio for the corresponding segment. This assessment assists us in the detection of the directions towards which the strongest—compared to the median average—events take place. Those occurrences indicate large increases in the zenith wet delay gradient that points in the direction of incoming moisture. Finally, we selected zenith wet delay gradient readings for the corresponding epochs, when the greater 75th percentile rainfall amount occurs. According to those epochs, we plotted the azimuthal distributions of the wet gradients to detect key directions. Similarly to the previous step, we used the stations UNSA, CAFJ, and PUNJ because they are complimented by precise rainfall data.

## 6. Results

In this section, we present the results for each analysis method following the same order as in Section 5.

### 6.1. k-Means Clustering of GNSS Time Series

Our k-means clustering analysis shows that the separation of the water vapour time series into three classes divides them according to their geographic position and seasonality (cf. Figure 4). The clustering is based on mean monthly time series and their seasonal gradients determine the cluster generation. The time series of the highest seasonal amplitude (cluster 3) shows an oscillating signal with a central value 27.5 mm and an amplitude of 12.5 mm. The baseline of cluster 2 is shifted by 10 to 15 mm. Cluster 1 represents the stations with the lowest seasonal amplitude.

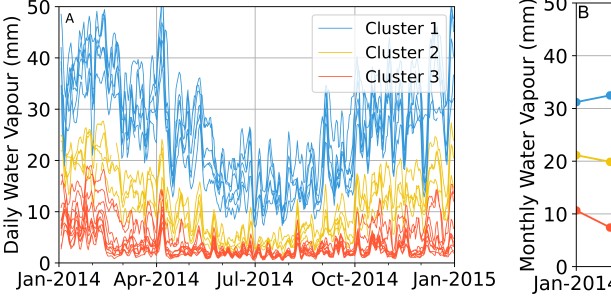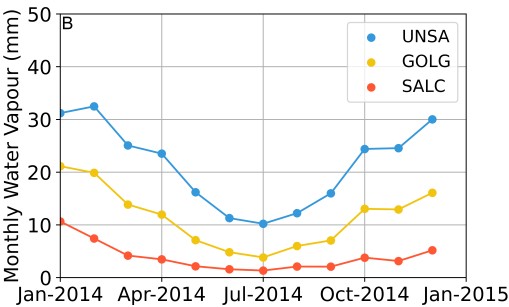

**Figure 4.** Results of clustering the water vapour readings showing all time series within a cluster with daily temporal resolution (**A**), but station clustering was performed on mean monthly values shown in (**B**). (**B**) shows individual mean monthly values along the topographic gradient exemplified for UNSA at 1224 m, GOLG at 2343 m, and SALC at 3799 m asl station elevation.

We observe the highest values in the austral summer during the SAMS seasons (January and February), whereas the lowest values take place in June. Moreover, October yields high magnitudes—in both the daily and mean monthly water vapour time series—followed by a slightly lower magnitude in November for 2014.

The map view of the station clusters show their expected geographic separation (cf. Figure 5). We emphasize that the clustering did not take into account the spatial location, but only the time series data. We separate stations by elevation and climatic conditions and use the 500 mm annual rainfall contour.

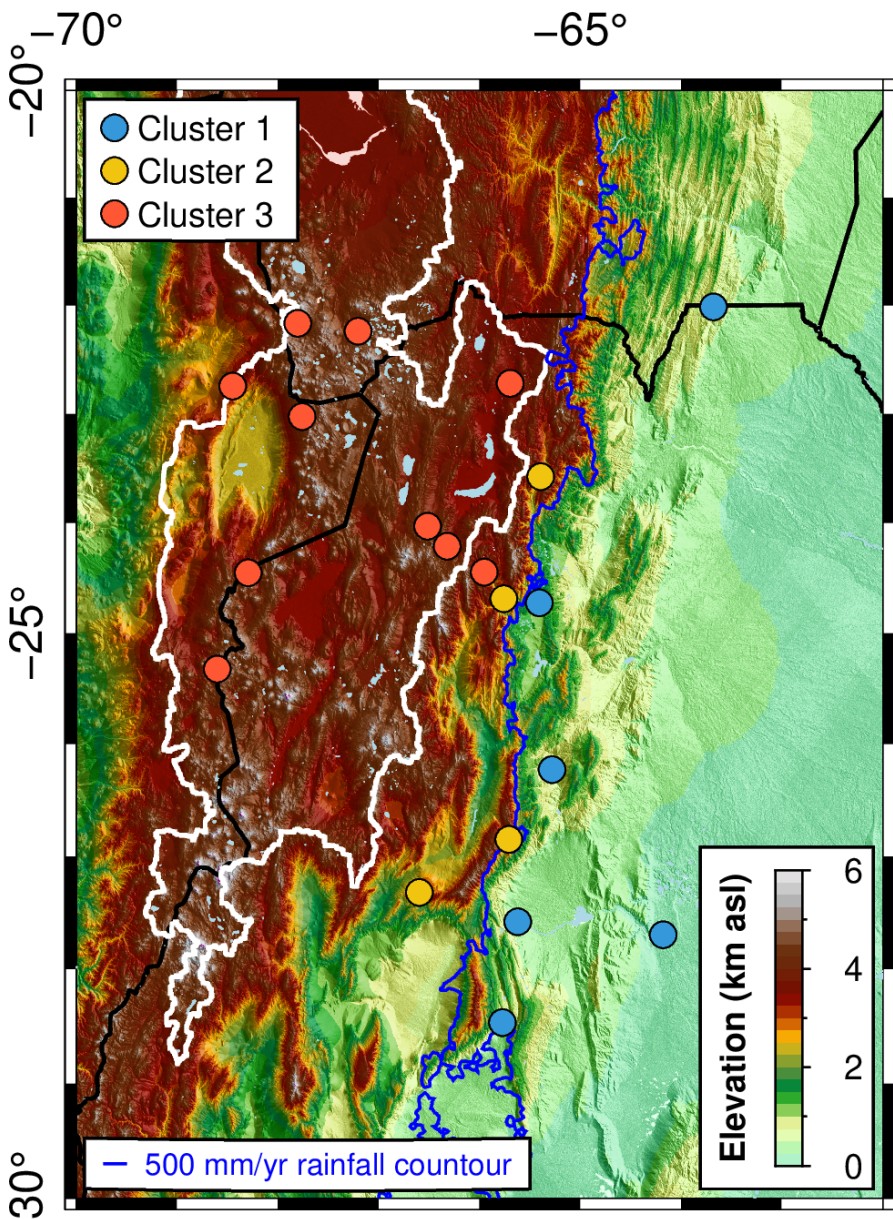

**Figure 5.** Map view of the clustered stations using the monthly mean values during 2014. Circle colours are similar to Figure 4. The stations with elevations up to 1225 m asl are located in the foothill zone (blue points), the sites with elevation between 2000 m and 2700 m asl are located in the transition zone (yellow points), and almost all stations above 3115 m asl are situated on the Altiplano–Puna plateau (red points). (Topographic data obtained from ETOPO1 [18], rainfall information retrieved from Tropical Rainfall Measuring Mission (TRMM) [20], as cited in Bookhagen and Strecker [10]).

*6.2. Spectral Analysis*

We analysed the spectrograms for all stations along the E-W and N-S directions (Figure 6). In our analysis, we only evaluated periods up to seven days, because this is the maximum range of the analysis window. The comparison between the UNSA, GOLG, SRSA, and SALC stations shows that the response strength is inversely proportional to the altitude. On the other hand, the stations along the N-S direction show a relatively homogeneous signal. Moreover, there is a notable difference in magnitude between the weaker winter and stronger summer signals. We also observe a diurnal signal independently of the station.

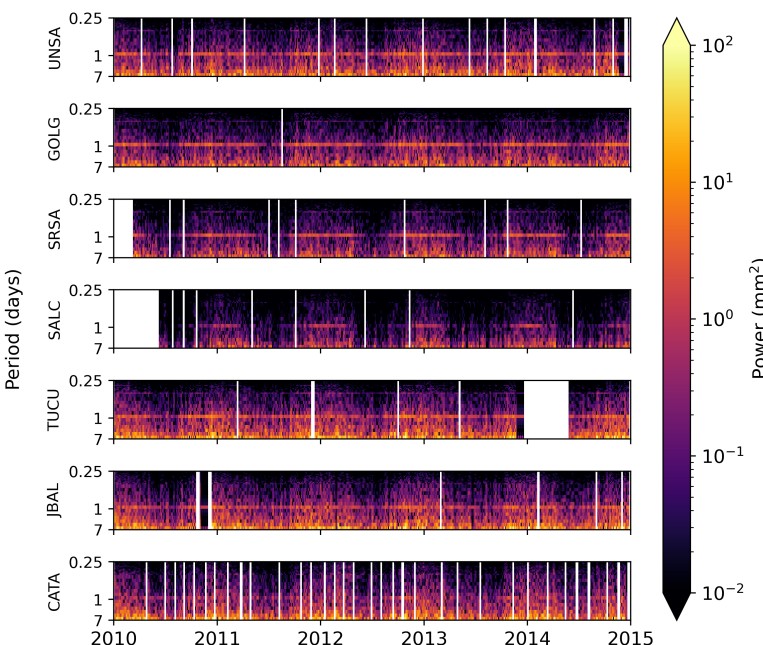

**Figure 6.** Spectral analysis of the stations along the E-W and N-S directions (Altitude asl: UNSA—1224 m, GOLG—2343 m, SRSA—3113 m, SALC—3799 m, TUCU—456 m, JBAL—381 m, and CATA—518 m; cf. Figure 1 and Table 1 for station location). A seven-day sampling window was used.

In Figure 7, we observe the PSDs of the UNSA and SALC stations. In addition, we highlight the power responses of the annual (An), semi-annual (S-An), monthly (M), weekly (W), and diurnal (D) periods. The strongest power signals at both stations are the annual, semi-annual, and diurnal periods. In a next step, we compare power signal strength between the stations: the higher elevations show significantly lower power for lower frequencies, in some cases, by one order of magnitude. The only exception is the half-year period, which is relatively higher in the SALC station.

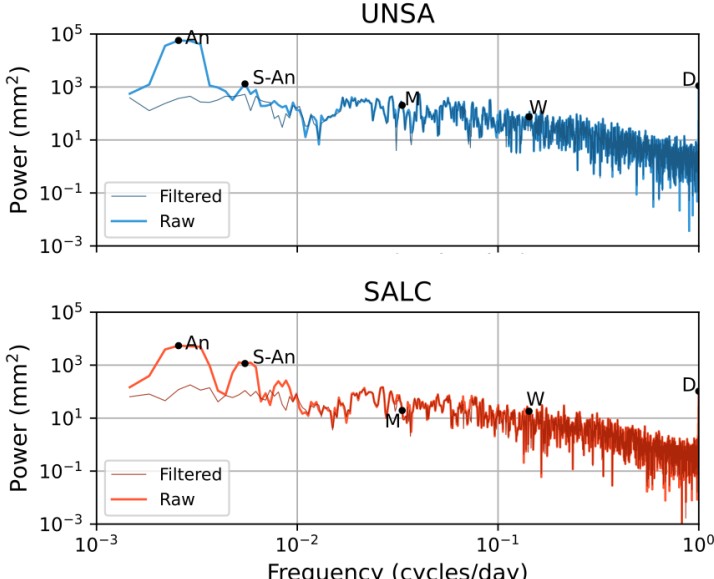

**Figure 7.** Power spectral densities (PSDs) of the low-elevation UNSA (upper panel) and the high-elevation SALC stations (lower panel). The thick and light-coloured lines represent the raw PSDs, while the thinner lines show the filtered values.

We note that the stations along the N-S cross-section are relatively homogeneous, while the stations along the topographic gradient alter significantly (Table 2). The decrease in the amplitude is at the level of 70–80%, which is in accordance with the observations made in Figure 6. In terms of the lower power signals, it appears that there is no physical relationship between those periods and the data series. Moreover, the normalized amplitudes show homogeneous behaviour for the daily and semi-annual signals, but we observe a decrease in the annual cycle at higher altitudes.

**Table 2.** Amplitude estimation of various harmonics for the stations along the E-W and N-S profiles.

| Period | Absolute and Std.-Normalized Amplitude (mm/—) | | | | | | |
|---|---|---|---|---|---|---|---|
| | UNSA | GOLG | SRSA | SALC | JBAL | TUCU | CATA |
| 1 day | 1.85/0.19 | 1.82/0.25 | 1.44/0.26 | 0.54/0.14 | 1.63/0.14 | 1.90/0.16 | 0.83/0.18 |
| 1 week | 0.30/0.03 | 0.08/0.01 | 0.05/0.01 | 0.11/0.03 | 0.37/0.03 | 0.54/0.05 | 0.53/0.05 |
| 1 month | 0.51/0.05 | 0.43/0,06 | 0.32/0.06 | 0.24/0.06 | 0.94/0.08 | 1.27/0.11 | 0.68/0.06 |
| 6 months | 10.13/1.01 | 7.22/1.00 | 5.39/0.97 | 3.12/0.80 | 11.77/0.99 | 11.73/0.98 | 10.16/0.96 |
| 1 year | 21.86/2.19 | 12.60/1.74 | 7.99/1.44 | 4.89/1.25 | 27.55/2.31 | 28.22/2.37 | 23.37/2.22 |

The power-law fitting of the binned filtered PSDs signals show a general agreement between the behaviours of the two stations (cf. Figure 8). The power-law exponents are comparable, but their roll-over magnitudes differ. With the exception of the low-magnitude spectrum, the log-binned datasets match well with this distribution. The estimated alpha values for UNSA and SALC stations are 1.50 and 1.46, respectively. Taking into account the power-law exponents, their standard deviations, and the degrees of freedom (18), the t-score equals 2.898. This value is lower than the threshold point of a t-distribution with a confidence interval of 99.9%, and we can assume that the two lines have comparable power-law slopes. Because of the different seasonal magnitudes of these two stations, we expect different roll-over magnitudes.

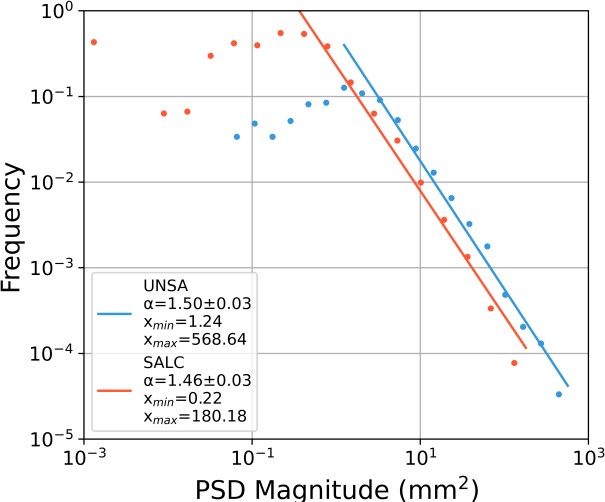

**Figure 8.** Power-law fitting of the filtered PSDs of the UNSA and SALC stations. The dots indicate the input data that are logarithmically binned using 20 classes. The power-law exponent is described by $\alpha$, and both exponents are comparable within their standard deviations. The fitting is constrained by the minimum value ($x_{min}$) that corresponds to the point/bin, where the frequency magnitude starts to decrease.

### 6.3. Water Vapour and Rainfall Relation

We analysed the differences between power-law fits and water vapour and rainfall amount (Figure 9). For our analysis, we filtered the greater than 90th percentile daily mean water vapour values, and we selected the cumulative rainfall readings for the corresponding

days. The water vapour series suggest that a power-law distribution is appropriate for modelling the log-binned observations. The exponent ($\alpha$), which describes the slope of the line, is similar for the low- and medium-elevation stations, but it changes significantly for the high-elevation stations on the Altiplano–Puna plateau. On the contrary, the maximum value ($x_{max}$), which is related to the shift of the slope on the x-axis, is negatively correlated with the altitude. With respect to the rainfall values, we observe that the differences between the slopes are less pronounced, and the PUNJ station shows a slightly steeper relation. This indicates a lower variation between the rainfall events in the UNSA station. Even though there is a trend of heavier events in lower altitudes, the CAFJ time series yields lower peak values than the higher-elevation PUNJ station. We observe that the standard deviations of power-law exponents for the water vapour and rainfall relation varies: exponent uncertainties are large for water vapour relation and uncertainties are reduced to 8–22% for the rainfall relation (Table 3).

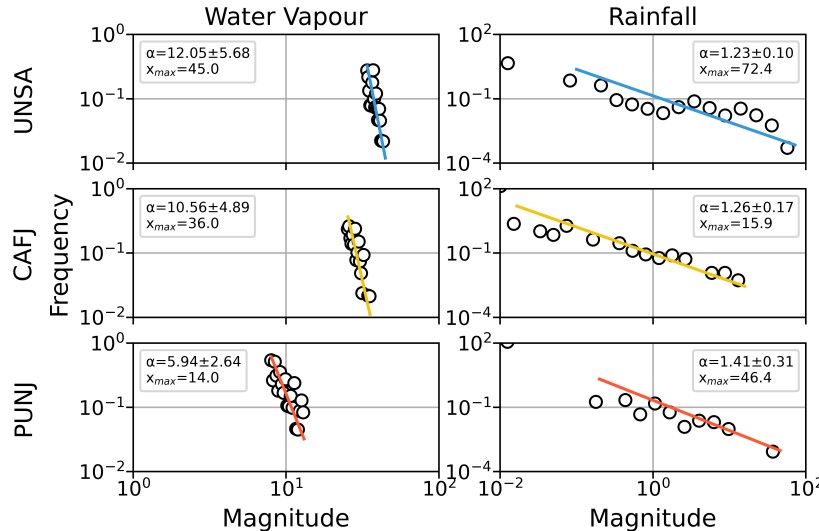

**Figure 9.** Power-law fitting of greater than the 90th percentile daily mean water vapour values (**left**) and the cumulative rainfall for the corresponding epochs (**right**) of the UNSA, CAFJ, and PUNJ stations. In both cases, the inputs (white dots) are grouped into 20 bins using a logarithmic scale. The minimum value ($x_{min}$) for the water vapour modelling was set to the lowest-magnitude bin, whereas the daily sums lower than 0.01 mm were omitted for the rainfall fitting.

**Table 3.** Statistical attributes for $\alpha$ and its standard deviation and $x_{max}$ of the power-law fitted lines in Figure 9.

| Station Name | Water Vapour | | Rainfall | |
| --- | --- | --- | --- | --- |
| | $\alpha$ | $x_{max}$ | $\alpha$ | $x_{max}$ |
| UNSA | $12.05 \pm 5.68$ | 45.0 | $1.23 \pm 0.10$ | 72.40 |
| CAFJ | $10.56 \pm 4.89$ | 36.0 | $1.26 \pm 0.17$ | 15.90 |
| PUNJ | $5.94 \pm 2.64$ | 14.0 | $1.41 \pm 0.31$ | 46.40 |

In the next step, we reverse the reference dataset, and we examine the relation between rainfall and their corresponding water vapour values. We exemplify this by selecting the 75th percentile cumulative rainfall events on daily basis and their corresponding mean water vapour values during those epochs. Taking into account the non-continuous presence of this scalar, we set a lower percentile threshold in order to retrieve sufficient data from all stations.

While we do not observe a relation between all rainfall events and water vapour amounts, we detect certain location-related tendencies when comparing stronger rainfall events and water vapour (cf. Figure 10). The scattering in the y-axis is broader in the

lower-elevation stations. On the other hand, the spreading in the x-axis follows the same trend but at a lower rate. However, the relations are very dynamic, and they cannot be described by a linear function.

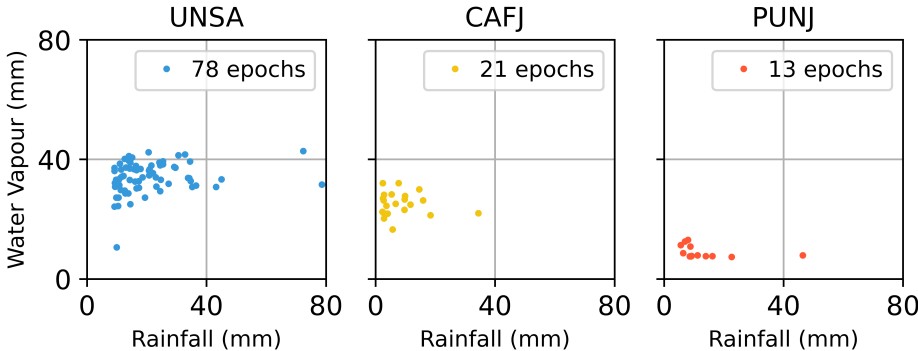

**Figure 10.** Quantile–quantile (Q-Q) plots of the greater than the 75th percentile daily-summed rainfall against the corresponding daily mean water vapour values of UNSA, CAFJ, and PUNJ stations. The lower number of measurements (epochs) in the higher-altitude stations is explained by the lower frequency of rainfall occurrences.

### 6.4. Zonal Moisture Gradient Transport

We compared the calculated zonal moisture gradient with ERA5-derived wind vectors (Figure 11). We observe annual wet gradients pointing in the general moisture directions to the north-west for UNSA and to the east for the high-elevation PUNJ station. This corresponds to the main wind direction for UNSA, which is toward the south-west. The observed eastward wind speeds in PUNJ are much higher, and they reflect the generally higher wind velocities following a high-to-low elevation gradient.

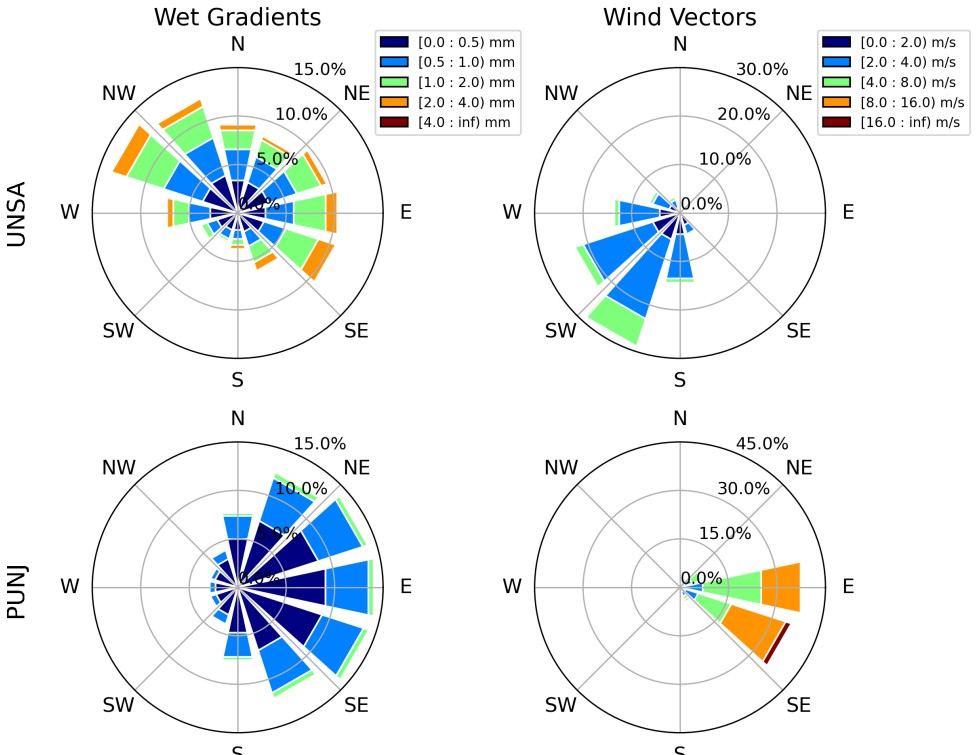

**Figure 11.** Azimuthal and magnitude distributions of the wet gradients and wind vectors of the UNSA and PUNJ stations. Wet gradients show the direction of moisture from the station location and

wind vectors show direction of wind transport. The azimuths are separated into 12 angular bins of 30°, and the magnitudes are arranged into five non-equal-width classes, in order to better illustrate distinct features for each station. The radius of the major influence is approximately 20 km. The wind directions are calculated from ERA5 hourly data on pressure levels [38] by extracting the median wind components between the station and the pressure level of 200 hPa. Because of the wind surface friction, the first km above the station is ignored. In both datasets we utilize a temporal resolution of 5 min, and the temporal coverage is adjusted to the shorter extent of the GNSS measurements.

We performed a seasonal analysis of wet gradient directions to highlight their strong seasonal dependence (Figure 12). For the low-elevation UNSA station located in the SALLJ we observe two peaks, and the majority of the azimuths point either to the east (90°) or to the west (270°) during austral spring/summer and fall/winter, respectively. During the fall season, there are higher values pointing to the west, which is reversed in the spring. The high-elevation PUNJ station shows different patterns: there is a peak azimuth direction from the north-east (45°) to the south-east (135°) throughout the year.

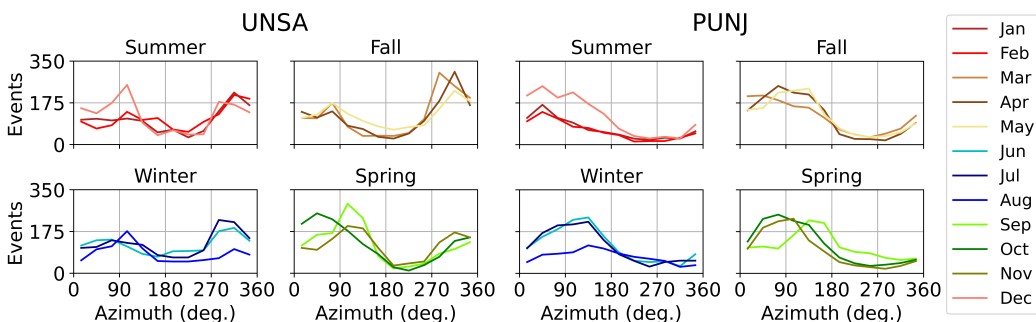

**Figure 12.** Azimuthal distribution of monthly-based wet gradients for the UNSA and PUNJ stations. Seasons are given as austral seasons (i.e., Summer is DJF), and the angular bins are defined as in Figure 11. The events indicate the number of hourly-sampled wet gradients that occur for each angular bin during each month.

Subsequently, we examined the ratios between the 90th to 50th percentile for all directions and seasons (Figure 13). In other words, we normalized the 90th percentile values by their medians and show the relation to the higher percentile: if the ratio is high, the 90th is much higher than the median. This analysis focuses on the wet gradient hotspots that are significantly larger than the mean, both in the spatial and temporal domain. Those occurrences are particularly interesting, because they indicate changing boundary conditions fluctuations. The ratios are high for the UNSA station and suggest a wide directional range. The PUNJ station shows a more homogeneous signal. There are no significant intra-seasonal variations of the distribution of the wet gradients in both cases. We also observe that moisture–gradient ratios vary by their directions. At the low-elevation UNSA station, the largest ratios occur during the summer and are directed towards the east-north-east (15° to 75° azimuth).

In the last step, we analysed the wet gradient direction during rainfall events exceeding the 75th percentile rainfall amount (Figure 14) in order to focus on those episodes. We observe that moisture gradients during rainfall events are spread out with a dominant direction from the south-east. In contrast, the high-elevation PUNJ station only shows five main direction that reflect local topographic shielding patterns and main moisture directions. We note that the majority of the events point towards the east in an azimuth range between 105–165° and 15–135° for UNSA and PUNJ, respectively.

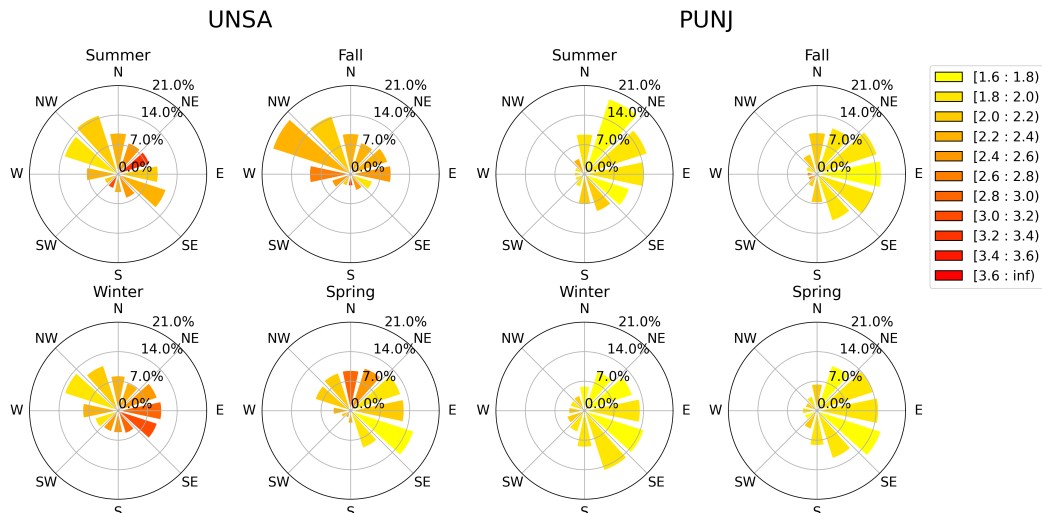

**Figure 13.** 90th to 50th percentile ratio of the wet gradients for the UNSA and PUNJ stations. The temporal and azimuthal separation is done similarly to Figures 11 and 12. The percentiles are extracted from the hourly averaged observations that occur in each directional and seasonal segment.

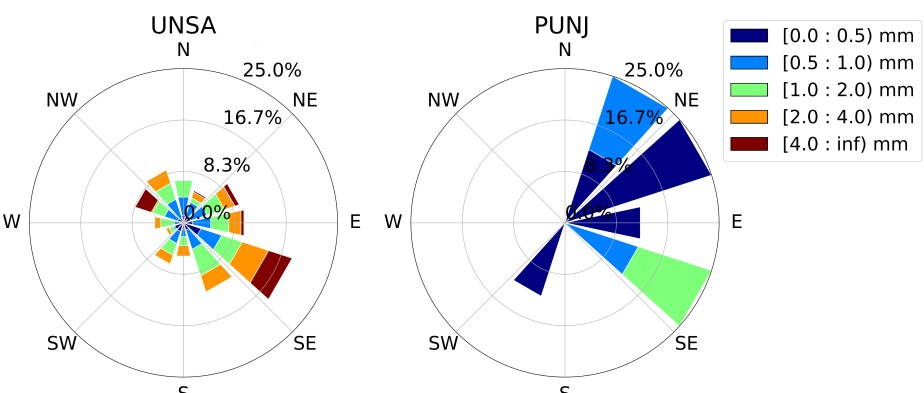

**Figure 14.** Summer season wet gradients for the larger than the 75th percentile rainfall events for the UNSA and PUNJ stations. The seasonal and azimuthal segmentation coincides with Figures 11–13, and the rainfall events are identified by the cumulative rainfall on 1 h basis that is measured with in situ rain gauges.

## 7. Discussion

The discussion follows the same organization of the Results sections.

### 7.1. k-Means Clustering of GNSS Time Series

We analysed the time series of the monthly water vapour during 2014 (Figure 4). The results of the clustering and the spatial water vapour distribution show the impact of topography and climate: the higher-elevation stations with low seasonal amplitudes are located on the arid Altiplano–Puna plateau, and the low-elevation densely vegetated areas are characterized by high seasonality. The transitional zone between these two end members shows an intermediate behaviour. This finding confirms our initial hypothesis and previous observations that orography plays a significant role on the local climate [10,32]. We further calculated the water vapour readings of UNSA at various altitudes, and we directly compare them with other stations (Figure 15). According to [83,84], the water vapour of a site can be projected to a higher altitude with the following equation:

$$wv = wv_0 \exp \frac{C_2 \Delta h}{1000} \tag{7}$$

where $C_2$ is a constant equal to 0.439, and $\Delta h$ is the height difference in meters.

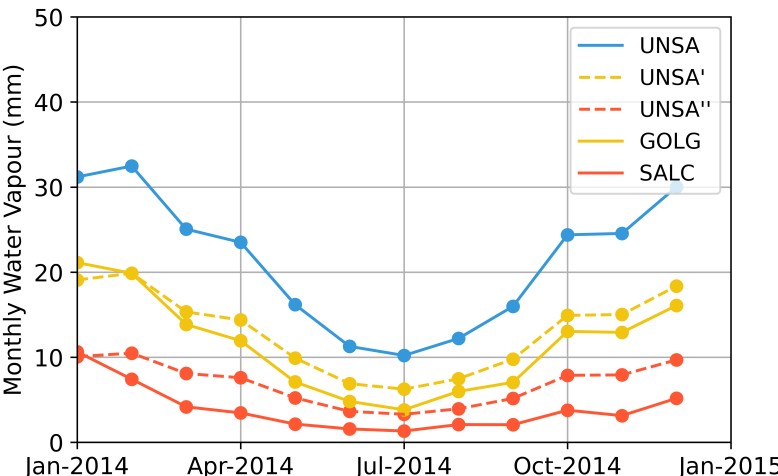

**Figure 15.** Projection of the mean monthly water vapour values of UNSA station (1224 m asl) at higher altitudes for the direct comparison of the readings with actual measurements at those points. UNSA' is the water vapour content at the corresponding height of the GOLG station at 2343 m asl, and UNSA" corresponds to the SALC station at 3799 m asl. In most cases, the water vapour contents at UNSA at the corresponding heights are higher than at the measured station.

Additionally, all stations show a seasonal signal, independent of altitude, and we observe a near-continuous signal from austral winter to spring (September to November). In order to examine this signal, we plot the monthly means over a longer period (Figure 16). The transitional seasons are characterized by larger 1 sigma standard deviations and show a larger variability in atmospheric water vapour.

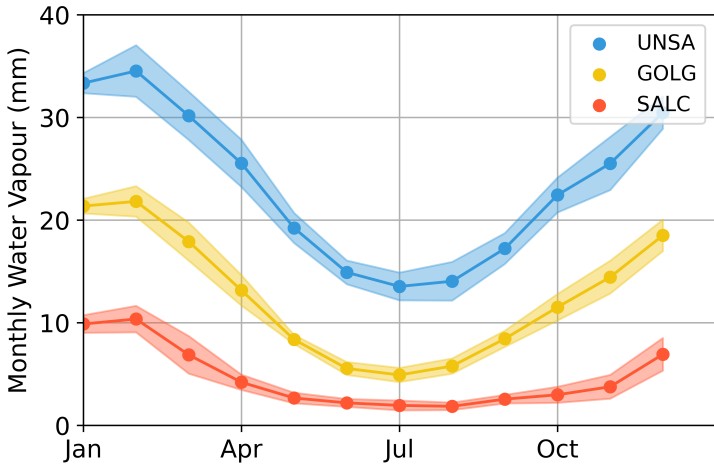

**Figure 16.** Mean monthly water vapour values for the stations along the topographic gradient from low to high elevations (cf. Figure 4). We analysed water vapour readings from 2010 to 2021 derived from ray tracing that only utilizes ERA5 meteorological data on pressure levels [38]. The semi-transparent colouring indicates the per-month standard deviation.

*7.2. Spectral Analysis*

We compare the signal responses at various frequencies of the GNSS-derived water vapour estimates against the ERA5 reanalysis data (Figure 17). We observe similar behaviour between the two spectrograms, which indicates high coincidence in the seasonal signals. In addition, the majority of the relative differences are less than 1%, showing that the GNSS observations are equally reliable. One advantage of GNSS data is that they can

achieve a significantly higher temporal resolution of five minutes, and detect features in this region on short time scales. In contrast, the reanalysis or mesoscale models have a temporal resolution of one hour and capture synoptic-scale processes.

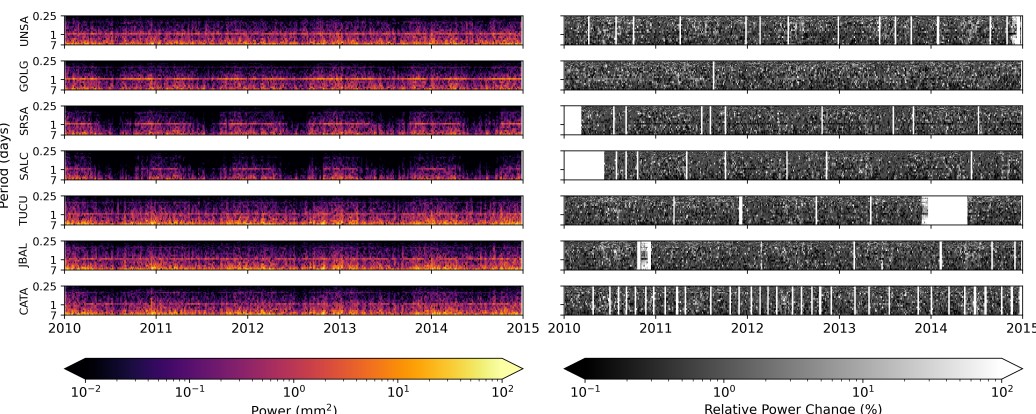

**Figure 17. Left:** Spectral analysis of the station along the E-W and N-S directions using a seven-day sampling window. This figure can be directly compared to Figure 6. In this case, we employed the ray-tracing derived water vapour using only meteorological information from ERA5 hourly data on pressure levels [38], instead of the GNSS-derived water vapour. **Right:** Difference in relative response power between the spectrograms of the GNSS- and the ERA5 ray-tracing derived water vapour series.

The spectral behaviour of the water vapour shows that all signals are primarily tuned at the annual, semi-annual, and diurnal periods, which correspond to the seasonal cycles of the moist air masses. The shape of the signal affects the ratio between period responses. In the case of the higher-elevation stations, the semi-annual periods are pronounced. This is due to the flattening of the water vapour series, which leads to a time series that can be better characterized by a harmonic equation with two oscillations. The water vapour values of the lower-elevation stations are reflected both in the spectrograms and the PSDs, showing the influence of the topography on the regional climate. Lastly, the frequency–magnitude relation of the filtered PSDs reveals relatively analogous responses to the seasonal fluctuations, regardless of location (cf. Figure 8).

*7.3. Water Vapour and Rainfall Relation*

The power-law fitting of the water vapour observations reveals higher values in lower altitudes on the one hand and a lower decay ratio for the PUNJ station. The prior is expected, and it has been noted in the previous sections (e.g., Figures 3 and 4). The latter is interesting, and we note that the high-percentile readings are more equally distributed in this station. Additionally, the fitting of the rainfall readings demonstrates that strong events will also take place at high elevations, but less frequently. However, this interpretation is not well applicable to the CAFJ station, because this station is part of an inter-mountain valley; thus the lower cloud coverage results in different temperature conditions. The high standard deviations of the exponents indicate skewed datasets that diverge significantly from a normal distribution. Despite the high uncertainty of the water vapour fitting, we observe clear trends. The Q-Q plots also show that extreme rainfall events are observed at high elevations, but they occur in a narrower water vapour peak range. This demonstrates the direct relationship between the required amount of water vapour to produce atmospheric saturation and the elevation. Additionally, the saturation is also dependent on the tropospheric temperature above the examined locations. In this case, there is a notable difference because of the within-the-layer temperature decrease along the altitude and the complex terrain that impacts cloud coverage.

*7.4. Zonal Moisture Gradient Transport*

We observe that the wind directions in the low-elevation stations are associated with the SALLJ and show moisture transport from the north-east and east towards the southerly directions (cf. Figure 11). Wind directions on the Altiplano–Puna plateau show only a minimal correlation with the SALLJ. There is a large difference in the wind speed between the low- and high-elevation stations, because the measurements do not take place at the same height. When considering a fixed pressure level (e.g., 500 hPa), the wind speed is homogeneous over the area. The distribution of the per-season-separated wet gradients shows dominant patterns for every station that slightly change through the year. This suggests a major influence of the topography and the altitude.

The higher 90th to 50th percentile ratios of the wet gradients in the low-elevation station are associated with the SALLJ that transports moist masses over the foothill zone of the south-central Andes. Moreover, the direction of the strongest occurrences (in terms of ratio) in this station (east-north-east) reveal important information. This is the direction of the topographic barrier which intersects with the SALLJ. The analogy between the wet gradients and the most intense rainfall occurrences suggests that most rainfall events reaching Salta are transported across the orographic barriers to the east and south-east of the city. In conjunction with our prior findings, this indicates good correlation between the wet gradients and the location of the wet air masses. Moreover, some rainfall events occur in the opposite direction, pointing to the orographic barrier west of Salta (cf. Figures 12 and 18). For the arid PUNJ station, we observe weaker wet gradients, but a strong correlation with topography. The correlation is pronounced because gradient generation is only triggered by the moisture transport on topographic barriers.

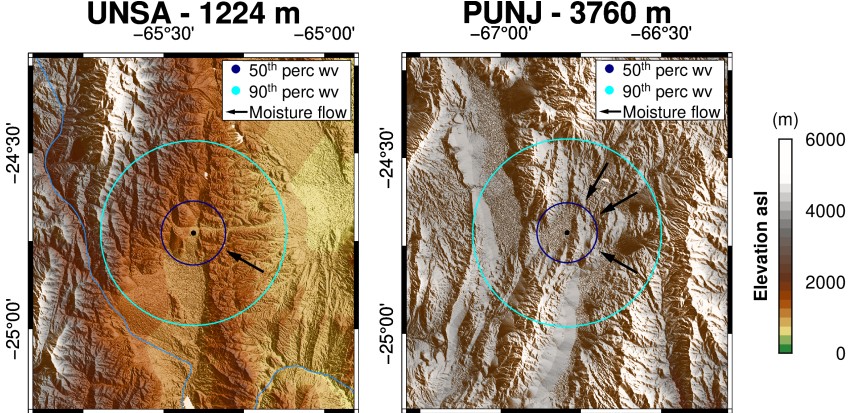

**Figure 18.** Topography setting of the low-elevation UNSA (1224 m asl) and high-elevation PUNJ (3760 m asl) stations. The dark- and light-blue circles show the area of influence of the 50th and 90th percentile of the water vapour above each site, respectively. The black vectors indicate the main moisture directions associated with high rainfall, as calculated in Figure 14. Topographic data obtained from ETOPO1 [18].

## 8. Conclusions

In this study, we have used GNSS time series data to better understand the climatic dynamics of the central Andes in northwestern Argentina. We have compiled 23 GNSS observations from 2001 to 2021 and have installed two GNSS stations that collected data between 2019 and 2021. The GNSS signal is used to measure water vapour content in the atmosphere at high temporal resolutions of 5 min. We use these data to make the following key observations:

(1)  The GNSS time series data show distinctive climatic behaviour for the Central Andes that was analysed using a clustering analysis: GNSS stations from the low-elevation area in the pathway of the SALLJ show similar behaviour and cluster into the same group. Stations from intermediate elevations at the transition from low- to high-

elevation areas show a distinctive signal and cluster into the same group. Stations from the high elevations located on the Altiplano–Puna plateau behave similarly and have the lowest absolute (not relative) seasonal component.

(2) A frequency analysis depicts the seasonal signals, and it illustrates the impact of the orographic uplift. The annual, semi-annual, and diurnal periods can be clearly identified, but there are also spectral differences across the time series. The most prominent variations between the stations are found in the magnitude of the water vapour levels, where the readings are inversely proportional to the station elevations.

(3) The association between water vapour and rainfall reveals a general correlation of stronger water vapour amounts. We observe that high water vapour episodes are less frequent at higher altitudes, but strong events still occur. We note that the rainfall–water vapour relation varies along the topographic gradient. At lower altitudes, rainfall occurs across a wide water vapour peak range. In contrast, at high elevations only a narrow band of water vapour amounts can be associated with rainfall events.

(4) We have used wet gradients to identify moisture transport for two sites: the low-elevation UNSA station at 1224 m and the high-elevation PUNJ station at 3760 m. The wet gradients allow us to document that local topographic effects strongly impact the characteristics of the GNSS and hydrologic stations. Even though the moisture fluxes' magnitude is subject to the circulation of the SALLJ and the mesoscale convective systems, nearby topography controls the circulation of atmospheric water vapour and controls the moisture pathways.

In comparison to reanalysis data, there are several advantages of GNSS meteorology techniques: a good accuracy of water vapour measurements, the ability to measure water vapour in three dimensions, and the high sampling rate of seconds to minutes. A network of homogenized and reliable GNSS stations will allow for an improved weather prediction. The National Geographic Institute of Argentina continuously operates a GNSS network that is very dense in urban zones, but lacks facilities in remote areas. Considering the ability of GNSS to measure the atmospheric moisture gradient, the aggregation of more stations and the integration of those data into meteorological applications would allow for short-term predictions of heavy rainfall, and would improve the weather prediction. This knowledge may help in reducing the damage from natural hazards, and it would benefit the agriculture sector, which is crucial for the local economy.

**Author Contributions:** Conceptualization, N.A., B.B. and K.B.; methodology, N.A. and B.B.; software, N.A. and K.B.; validation, N.A. and B.B.; formal analysis, N.A., K.B. and G.D.; investigation, N.A., B.B. and K.B.; resources, J.W. and B.B.; data curation, N.A.; writing—original draft preparation, N.A.; writing—review and editing, B.B., J.W., K.B., A.d.l.T. and G.D.; visualization, N.A. and B.B.; supervision, B.B., J.W. and A.d.l.T.; project administration, B.B. and J.W.; funding acquisition, B.B. and J.W. All authors have read and agreed to the published version of the manuscript.

**Funding:** This research was funded by the German Research Foundation (Deutsche Forschungsgemeinschaft—DFG) and the federal state of Brandenburg through the International Research Training Group-StRATEGy (DFG IRTG2018). Kyriakos Balidakis is funded by the DFG through the TerraQ Project (ID 434617780—SFB 1464). The publication costs were covered by the DFG within the funding programme "Open-Access-Publikationskosten" and the German Research Centre for Geosciences (Deutsches GeoForschungsZentrum—GFZ).

**Data Availability Statement:** The ETOPO1 dataset was provided by the National Oceanic and Atmospheric Administration (NOAA). The MODIS/Terra dataset was provided by the National Aeronautics and Space Administration (NASA). The Tropical Rainfall Measuring Mission (TRMM) dataset was provided by NASA and the Japanese Aerospace Exploration Agency (JAXA). The ERA5 dataset was provided by the European Centre for Medium-Range Weather Forecasts (ECMWF). The water vapour estimates and the zenith total delays, as well as their gradients, were provided by the German Research Centre for Geosciences (Deutsches GeoForschungsZentrum—GFZ) and the Nevada Geodetic Laboratory (NGL) (http://geodesy.unr.edu/, accessed on 3 December 2020). The entire time series of the GNSS and ray-tracing products are provided by the GFZ Data Services (https://dataservices.gfz-potsdam.de/panmetaworks/showshort.php?id=8fd77904-2f86-11

ed-9732-32bb1430b4f7, accessed on 9 September 2022). Precise rainfall information was provided by an in situ rain-gauge station, the Nationa Institute of Agriculture Technology (Instituto Nacional de Tecnología Agropecuaria—INTA) of Argentina (http://siga.inta.gob.ar/, accessed on 16 December 2021), and Meteostat (https://meteostat.net, accessed on 16 December 2021).

**Acknowledgments:** We would like to thank Fernando Hongn and Luis Alvarado from the Institute of Bio- and Geosciences of Northeast Argentina (Instituto de Bio y Geociencias del NOA—IBIGEO) for their assistance in installing the ground stations.

**Conflicts of Interest:** The authors declare no conflict of interest.

## Abbreviations

The following abbreviations are used in this manuscript:

| | |
|---|---|
| EPOS | Earth Parameter and Orbit System |
| EVI | enhanced vegetation index |
| E-W | east–west |
| GFZ | Deutsches GeoForschungsZentrum (German Research Centre for Geosciences—ger) |
| GNSS | global navigation satellite system |
| GPS | global positioning system |
| IGN | Instituto Geográfico Nacional (National Geographic Institute of Argentina—sp) |
| NGL | Nevada Geodetic Laboratory |
| N-S | north–south |
| PDF | probability density function |
| PSD | power spectral density |
| Q-Q | quantile–quantile |
| SALLJ | South American low-level jet |
| SAMS | South American monsoon system |
| UP | University of Potsdam |

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
