# Peer review of "Water-Vapour Monitoring from Ground-Based GNSS Observations in Northwestern Argentina"

_remotesensing, doi:10.3390/rs14215427_

Round 1

Reviewer 1 Report

For a detailed description of the advantages of GNSS meteorology, it is necessary:

- to study the vertical profiles of humidity using the approach of tropospheric tomography.

- having troposphere research data for the period 12 yeras, it is desirable to study the possibility of using GNSS meteorological data for climate change research,

- the article claims that GNSS meteorology will improve weather forecasting. These statements must be supported by specific facts.

- closer cooperation with meteorologists.

Author Response

Dear reviewer, thank you very much for taking your time to review our paper. The focus of our publication is primarily the understanding of the water-vapour circulation over northwestern Argentina. Additionally, our secondary goal is to demonstrate the GNSS capabilities with an approximation that is not often followed. Instead of only using the Zenith Wet Delays, we also incorporate in our investigation their gradients.

According to the guidelines of the journal, we have rewritten the results and discussion sections of this manuscript and have modified other parts of the text.

We would like to point out that the density of the network in this area is sparse and it is not homogenously distributed across space and elevation, and the network does not allow for a calculation of a tropospheric tomography. Although the time span of our data is 12 years, there are only a few stations that cover the entire period, and most importantly, all of them are located in low altitudes. As for the incorporation of the GNSS data into meteorological applications, in the introduction we show this possibility, and we cite three references (doi: 10.5194/amt-12-345-2019, 10.1038/s41598-017-12593-z, and 10.1785/0220170162).

In our “Conclusions” section, we modified the text according to your comments, stating that the responsible agency for the GNSS network must aggregate more stations in remote areas. Additionally, the national meteorological service should incorporate those measurements for nowcasting and weather prediction, which is not yet done for this area. Considering the results of this paper and the non-use of the data, it would be beneficial to examine the influence of those measurements in cooperation with meteorologists in a further step. We emphasize that one of the co-authors of this study (A. de la Torre) has been actively involved in generating improved weather-prediction models for the area around Mendoza to the south of our study region.

Reviewer 2 Report

The authors tried to quantify the dynamics of water vapour transport by analyzing time series of precipitable water vapour from the dual frequency GNSS observation network in northwestern Argentina, complemented by in situ rainfall measurements and ERA5 reanalysis data. Their results show a strong relationship between altitude and the water-vapor content, as well as between the transport pathways and the topography. The results would certainly help in various ways to the local economy and holds a great importance. Moreover, the authors have presented well with appropriate methodology, analysis, and discussions. 

Author Response

Dear reviewer, thank you very much for taking your time to review our paper. Your recognition of our publication is well appreciated. We have considered the comments of all reviewers and provide an improved version of our work. Additionally, we have rewritten the results and discussion sections of this manuscript and have modified other parts of the text, according to the guidelines of the journal.

Reviewer 3 Report

Remote Sensing - 1943301

Water-vapour monitoring from ground-based GNSS observations in northwestern Argentina

By Nikolaos Antonoglou, Kyriakos Balidakis, Jens Wickert, Galina Dick, Alejandro de la Torre and Bodo Bookhagen

===================================================================

General comments:

This paper uses long term GNSS stations that monitor the larger Argentinian Andes region both with regards to its elevation variations and its North-South extension.

The paper is very well written and precise in describing the methodology followed.

The detailed analysis of the GNSS water vapor signal with respect to its E-W and N-S gradients as well as the use of ERA5 reanalysis for dynamics fields correlations provide a very serious and precise plan of analysis.

Likewise, the methodology developed is sound and quite adapted to the questions raised in the article. Its presentation is also very clear and precise.

However, the outcome falls somewhat short of providing more than what is already well known in the literature: IWV varies with altitude but is also influenced by water vapor transport linked to the relevant orography context.

Furthermore, comparing the variability of IWV (amplitude of seasonal cycles, range of variations, …) of stations at significantly different altitudes on the value of their absolute values rather than their relative variation is highly questionable, but can also led to conclusions that carry little information.

Hence, this paper has potential for significant results if it can sort out water vapor value thresholds or patterns together with specific meteorological regimes which will lead to high impact heavy precipitations, in order to make this work a step forward towards decision tools for improved forecasting and, thus, people and territories safeguard.

Specific comments:

L 109:             I believe that is “Table 1”.

L 126-136:      All the sited stations should be clearly identified on Figure 1.

L 138:             Ok for the time span... but the period of interest is still 2010-2019!

L 264:             Clarify the use of the “forecasted” data... is this some advanced interpolation? This should be detailed beyond the mere reference to the Prophet algorithm to which all readers are not necessarily accustomed to.

L 300-301:      “when water vapor reaches the peak relative humidity”.. obtained from which source of measurement… GNSS provides column water vapor, not relative humidity. Using IWV to make the selection of events means including possible seasonality bias in the data set!

L 303:             “90th percentile” ... of what ? IWV ?... that can’t be, see above comment.

L 337-340:      that is known as the altitude effect, i.e., there is less water vapor aloft!

L 351-353:      Precise exactly which month show “larger 1 sigma standard deviations” and if there are differences between data sets.

L 354-359:      Discussion based on absolute values is somewhat biased as it is well know that total IWV is altitude dependent. Cycle amplitudes at each altitude levels are quite similar in relative values. Actually, it is even higher at higher elevations and lower at lower elevations, in opposite with your argument!

L 437-439:      Once again, the relative scattering is similar if one accounts for the water vapor altitude dependence.

L 440-448:      The interpretation holds well between UNSA and PUNJ… but not with CAFJ. This must question the rain regime effects.

L 448-450:      The narrower peak range is due to the fact that there is less water vapor at higher altitude, but saturated conditions can still hold! I don’t think there is nothing new to that.

L 453-454:      Isn’t water vapor expected to be advected by wind, bringing high humidity air masses to support the orography effect. Thus, a broader view of the atmospheric dynamics is needed to understand this result.

L 461:             during austral spring/summer and fall/winter, respectively.

L 465-475:      This section is a bit hard to follow and does lead to a clear or conclusive statement.

L 474-475:      The statement made is correct but the figure does help supporting this easily by eye. Indeed, one has to understand that the “like-colors” color code is more important that the size of the directional wedges in the interpretation. Maybe a color code that would bring the results more “ti life” could be helpful.

L 465-475:      As a consequence of the above, a discussion about the spatial distribution (the size of the wedges) is missing.

L 481-482:      This statement stands for PUNJ, but for UNSA is looks more like 105 to 145, is that right?

L 487:             Wind speed/direction isn’t shown or even presented before the statement!

L 493:             the strongest occurrences” … be sure to clearly indicate that this is with respect to percentile ratio values in contrast to % of occurrences!

Figure 17:       In the caption, indicate “the main moisture direction associated with high precipitation” to be more precise in the statement.

L 515:             “the lowest absolute (not relative) seasonal component”

L 518-520:      Nothing ground breaking with that… that has been known for at least 2 decades!

L 521-522:      The same could almost be said for this statement either.

L522-523:       Actually, high altitude water vapor conditions associated the local heavy precipitations seem to show a different behavior, and that is interesting and should be investigated on why this happen?

L 524-526:      Again, if you think relative rather than absolute…! Moreover, the reflection should rather be on the water vapor needed at a given altitude to provide saturation.

L 528:             Name of the stations should be clearly associated with its 4-character code here, rather than having to search the map.

L 530-531:      What about SALLJ circulation?

L 534-535:      This sentence is highly anticipated with regards to the current work as it does not even offer some “decision tree” suggesting that for such or such conditions of IWV values associated with such wind direction and/or amplitude, then high precipitations are expected.

Conclusions:

This paper is very well written and offer a very detailed and in-depth analysis of the GNSS data from different stations within a large network that documents water vapor along and aggress the Andes.
Nevertheless, as it stands it lacks true scientific insights beyond the logical statement that water vapor varies with altitude and orography can play a role to water vapor distribution variability.

Likewise, the discussion on possible links with heavy precipitation does not offer any conclusions beyond statistics.

Hence, this paper should not publish before more cautious and in-depth analysis is perform to offer what seems to be the drive of the work in the introduction: a way to improve heavy precipitation forecasting.

Author Response

Dear reviewer, thank you very much for taking your time to review our paper. The focus of our publication is primarily the understanding of the water-vapour circulation over northwestern Argentina. Our secondary goal is to demonstrate the GNSS capabilities with an approximation that is not often followed. Instead of only using the Zenith Wet Delays, we also incorporate in our investigation their gradients. We have considered the comments of all reviewers and provide an improved version of our work. Additionally, we have rewritten the results and discussion sections of this manuscript and have modified other parts of the text, according to the guidelines of the journal. In the following lines, we address all your specific comments. In the case the observations are connected to each other, we provide answers for the block of comments.

L 109:             I believe that is “Table 1”.

This has been fixed.

L 126-136:      All the sited stations should be clearly identified on Figure 1.

The caption has been corrected.

L 138:             Ok for the time span... but the period of interest is still 2010-2019!

This has been fixed.

L 264:             Clarify the use of the “forecasted” data... is this some advanced interpolation? This should be detailed beyond the mere reference to the Prophet algorithm to which all readers are not necessarily accustomed to.

We have given more explanation about the algorithm.

L 300-301:      “when water vapor reaches the peak relative humidity”.. obtained from which source of measurement… GNSS provides column water vapor, not relative humidity. Using IWV to make the selection of events means including possible seasonality bias in the data set!

L 303:             “90th percentile” ... of what ? IWV ?... that can’t be, see above comment.

In this case, we do not use relative humidity measurements. This statement explains the theory regarding the generation of precipitation, and its relation to water vapour. In other words, it says that the occurrence of a precipitation event requires saturation of a part of the atmosphere (relative humidity 100%), which leads to water condensation. The water-vapour seasonality cannot be examined simultaneously in all stations because the length of the observations is not sufficient. This issue is also mentioned in the calculation of the power-spectral densities in the “Spectral Analysis” section.

L 337-340:      that is known as the altitude effect, i.e., there is less water vapor aloft!

L 354-359:      Discussion based on absolute values is somewhat biased as it is well know that total IWV is altitude dependent. Cycle amplitudes at each altitude levels are quite similar in relative values. Actually, it is even higher at higher elevations and lower at lower elevations, in opposite with your argument!

Apart from the gravity, there are various factors that influence the water vapour level. A second factor is the atmospheric temperature. More specifically, at 3000 m asl altitude the temperature is significantly lower, and thus the water-vapour holding capacity of the air molecules is also lower. Additionally, the topography is another factor that impacts the local climate because it influences the cloud coverage and local circulation systems and wind velocities. This statement is also analyzed in the comment regarding lines 440-448.

L 437-439:      Once again, the relative scattering is similar if one accounts for the water vapor altitude dependence.

In this case, we point out that the effect is more evident because of the orography. In order to prove this, we project the water-vapour values of a low-altitude station to higher levels and compare them with station measurements (cf. Figure 15).

L 351-353:      Precise exactly which month show “larger 1 sigma standard deviations” and if there are differences between data sets.

In Figure 4 we show the monthly mean water vapour values. In the series, we mention that the gradual advancement of the water vapour is broken because the values in November are lower than in October. In order to examine if this is a systematic effect, we plot in Figure 16 the monthly means of a longer period along with their 1-sigma standard deviations. The averaged series show that the fluctuation from September to November (also referred as transitional seasons in the text) is due to climate variability.

L 440-448:      The interpretation holds well between UNSA and PUNJ… but not with CAFJ. This must question the rain regime effects.

Indeed, the interpretation does not hold well with CAFJ. This station is part of an intermontane valley; thus, the lower cloud coverage and higher local heating lead to heat-up that does not take place in UNSA. We have added some comments to the “Discussion” part.

L 448-450:      The narrower peak range is due to the fact that there is less water vapor at higher altitude, but saturated conditions can still hold! I don’t think there is nothing new to that.

It is maybe nothing new to that theoretical statement, but there is no similar investigation for this area. Additionally, this is not our final conclusion; it is a good step to prove the theory and continue to the next experiment.

L 453-454:      Isn’t water vapor expected to be advected by wind, bringing high humidity air masses to support the orography effect. Thus, a broader view of the atmospheric dynamics is needed to understand this result.

The description of the orographic process is done in the initial overview of the manuscript. It is just the first phase and we do not use the outcome to directly derive final conclusions.

L 461:             … during austral spring/summer and fall/winter, respectively.

This has been fixed.

L 465-475:      This section is a bit hard to follow and does lead to a clear or conclusive statement.

In this section, we examine the per-direction dynamics for each section. In order to do this, we calculate the 90th to 50th percentile ratios of the gradients. Low ratios (also referred as homogeneous stations in the text) do not indicate changes in the boundary conditions, while the opposite stands for high ratios.

L 474-475:      The statement made is correct but the figure does help supporting this easily by eye. Indeed, one has to understand that the “like-colors” color code is more important that the size of the directional wedges in the interpretation. Maybe a color code that would bring the results more “ti life” could be helpful.

The colourmap has been changed.

L 465-475:      As a consequence of the above, a discussion about the spatial distribution (the size of the wedges) is missing.

The size of the wedges depends on the aperture angle (30o in this case) and the radius that influences the GNSS observations. Considering an elevation angle of 7 o, this radius is about 20 km. This is shown in Figure 18.  We have added some comments to the caption.

L 481-482:      This statement stands for PUNJ, but for UNSA is looks more like 105 to 145, is that right?

This has been fixed.

L 487:             Wind speed/direction isn’t shown or even presented before the statement!

Here we discuss the results related to Figure 11, the legend includes wind speed information, and the azimuthal distribution is given by the roses. In the caption, we explain our argument, where the wind measurements are taken. In order to be more clear, we direct the text to the figure.

L 493:             “the strongest occurrences” … be sure to clearly indicate that this is with respect to percentile ratio values in contrast to % of occurrences!

This has been fixed.

Figure 17:       In the caption, indicate “the main moisture direction associated with high precipitation” to be more precise in the statement.

This is fixed. Precipitation is more general, this area experiences only rainfall. Thus, we only use this term. The figure number changed due to section reordering.

L 515:             “the lowest absolute (not relative) seasonal component”

This has been fixed.

L 518-520:      Nothing ground breaking with that… that has been known for at least 2 decades!

As stated in previous comments, we do not only examine the decrease of the water vapour with the elevation, but we correlate it with the influence of the mountain range. We have enriched the discussion part of the first experiment, and we have added proof.

L 521-522:      The same could almost be said for this statement either.

This sentence does not stand alone as a finding, it is complementary to the paragraph. Even though it is nothing ground breaking, the entire statement gives information about the area which are not observed in other publications.

L522-523:       Actually, high altitude water vapor conditions associated the local heavy precipitations seem to show a different behavior, and that is interesting and should be investigated on why this happen?

L 524-526:      Again, if you think relative rather than absolute…! Moreover, the reflection should rather be on the water vapor needed at a given altitude to provide saturation.

The comparison is absolute because the relation with the altitude is not straightforward, and we cannot just make a normalization. The relative comparison was held when we compared the water-vapour decay for various locations. In general, saturation occurs either by a decrease in the temperature or by an excessive amount of water vapour. The different behaviours occur due to the significantly different amounts of water vapour and the different temperature profiles in the area. As for the latter, we have already mentioned that the tropospheric temperature above the examined locations differs because of the temperature drop along the altitude (within this layer). We have added some comments to the  “Discussion” part. 

L 528:             Name of the stations should be clearly associated with its 4-character code here, rather than having to search the map.

This has been fixed.

L 530-531:      What about SALLJ circulation?

We have added some comments to the “Conclusions” part, mentioning the circulation of the SALLJ and the Mesoscale Convective Systems. Additionally, those two parameters are already described in the “Introduction” part.

L 534-535:      This sentence is highly anticipated with regards to the current work as it does not even offer some “decision tree” suggesting that for such or such conditions of IWV values associated with such wind direction and/or amplitude, then high precipitations are expected.

We have added some comments to the “Conclusions” part, pointing out that the GNSS data are not used at all in meteorological applications, and suggesting the exploitation for nowcasting and weather prediction.

Round 2

Reviewer 3 Report

The authors have provided the requested improvements for clarity and precisemness of the manuscript as well as conclusive arguments to make their points hold.

Givent this paper also presents observations never reported with such in-depth analysis over the given geographical area, it is of significant interest to the scientific community.

Finally, the manuscript is well written and the methodology and analysis are clearly presented.

Hence, the paper is recommanded for publication as is without any further review.